# Using Inertial Measurement Units to Examine Selected Joint Kinematics in a Road Cycling Sprint: A Series of Single Cases

**DOI:** 10.3390/s24113453

**Published:** 2024-05-27

**Authors:** Simon Morbey, Marius Tronslien, Kunho Kong, Dale W. Chapman, Kevin Netto

**Affiliations:** Curtin School of Allied Health, Curtin University, Kent Street, Bentley, WA 6102, Australia; simon.r.morbey@gmail.com (S.M.); mtronslien@gmail.com (M.T.); cedric.kong93@gmail.com (K.K.); dale.chapman@curtin.edu.au (D.W.C.)

**Keywords:** sprint cycling, road cycling, inertial measurement units, kinematics, elite

## Abstract

Sprinting plays a significant role in determining the results of road cycling races worldwide. However, currently, there is a lack of systematic research into the kinematics of sprint cycling, especially in an outdoor, environmentally valid setting. This study aimed to describe selected joint kinematics during a cycling sprint outdoors. Three participants were recorded sprinting over 60 meters in both standing and seated sprinting positions on an outdoor course with a baseline condition of seated cycling at 20 km/h. The participants were recorded using array-based inertial measurement units to collect joint excursions of the upper and lower limbs including the trunk. A high-rate GPS unit was used to record velocity during each recorded condition. Kinematic data were analyzed in a similar fashion to running gait, where multiple pedal strokes were identified, delineated, and averaged to form a representative (average ± SD) waveform. Participants maintained stable kinematics in most joints studied during the baseline condition, but variations in ranges of movement were recorded during seated and standing sprinting. Discernable patterns started to emerge for several kinematic profiles during standing sprinting. Alternate sprinting strategies emerged between participants and bilateral asymmetries were also recorded in the individuals tested. This approach to studying road cycling holds substantial potential for researchers wishing to explore this sport.

## 1. Introduction

In competitive cycling, sprinting is an essential component that determines the result of many races. In the three Men’s Grand Tours, The Giro d’Italia, Tour de France, and Vuelta a España, one in every three stages are decided by a mass, small bunch, or a head-to-head sprint [1,2,3]. A sprint is defined by a sudden increase in power output and effort leading to a sustained acceleration [1]. This can happen in the closing meters of the race as competitors attempt to be the first cyclist across the finish line [2]. In this situation, the sprints are most commonly completed in an out-of-saddle (standing) position; however, for longer sprints or sustained attacks during a particular stage, these sprint-type efforts may be completed in an in saddle (seated) position. Menaspà and colleagues [2] determined that sprints in the men’s races last an average of 13 s in duration, with an average speed of 64 km/h, while Peiffer et al. [4] reported that sprints in women’s races typically last 22 s, with an average speed of 54 km/h. Despite being such an important component in competitive cycling, there is a paucity of information concerning the kinematics involved in the action of sprinting.

Most research investigating biomechanics in road cycling has been performed in a controlled laboratory setting and does not focus on the action of sprinting. For example, Bertucci et al. [5] compared the difference in crank torque and the rate of perceived exertion (RPE) of riding on an ergometer in a lab against riding a bicycle outdoors. They observed notable differences in both variables, with increases recorded for values collected on the ergometer. This is believed to be due to an increased stiffness and damping of forces on the ergometer, as well as the fact that the rider is overcoming the load of the flywheel on the ergometer opposed to their own mass on the bicycle outdoors [6]. Changes in environment (laboratory vs. outdoors) are likely to influence the cyclist’s biomechanics during sprint cycling, particularly through the upper limbs, as the front steering mechanism of a bicycle is not fixed, unlike an ergometer [6]. Further, a systematic review by Johnston et al. [7] investigating cycling knee biomechanics reported that all 14 studies included were completed in a controlled laboratory on either a stationary ergometer or stationary bicycle. These researchers drew similar conclusions to Fregly et al. [6] and Burnie et al. [8], who highlighted the limitations in knowledge when a biomechanical analysis is not conducted in an environment representative of that in which the sport is competed [7]. 

Previously, Costes et al. [9] reported that the upper limb transfers 3–5% of energy into total crank power output during cycling. This upper limb function becomes more critical with high-intensity cycling, in which the power output is reduced to 10–20% if upward forces are not produced on the handlebar [10]. During seated cycling, the energy produced by muscles is delivered into three points: pedals, seat, and handlebars [11]. It has been hypothesized the upper limb joints stabilize the trunk as well as creating seat force and transferring energy down to the lower limb. An increase in the force produced by the legs leads to an increase in the upward force at the hip joint. This force production results in a decrease in the seat reaction force, which requires cyclists to compensate by generating upper-limb force to pull on the handlebar until they can reach an adoption point [12]. Moreover, alternate bilateral movements of the left and right leg result in decreased trunk stabilization, which results in the upper limb absorbing extra force to increase stability during cycling [9]. These findings are derived from an understanding of the mechanical movements in cycling. However, no empirical kinematic data support these hypotheses. Thus, this research aimed to describe selected joint kinematics during a cycling sprint outdoors. We sought to evaluate two different types of cycling sprints: standing and seated sprinting, comparing these to a seated baseline condition of cycling at 20 km/h. We also piloted a running gait analysis approach to process the cycling kinematics, allowing insights from multiple pedal revolutions to be assessed. 

## 2. Materials and Methods

We recruited three participants for this cross-section, case series study design. Each participant was judged to be “Tier 3” or “Highly Trained/National Level” based on the 6-tier system proposed by McKay et al. [13]. Participant 1, who is female (age: 21 years, weight: 56 kg, height: 1.55 m), competes at an elite and U23 level in international triathlon competition and is ranked in the Top 500 in the World Triathlon Rankings and the Top 70 in the Asian Triathlon Rankings. Participant 2, who is male (age: 20 years, weight: 69 kg, height: 1.79 m), competes at a U23 level in international triathlon competition and is ranked in the Top 15 of the U23 Triathlon Australia Rankings. Participant 3, who is male (age: 19 years, weight: 76 kg, height: 1.87 m) competes at a U23 level in international triathlon competition and is ranked in the Top 15 of the U23 Triathlon Australia Rankings. Although these participants were from a triathlon background, they had predominantly competed in draft-legal races more akin to cycling criterium racing. They were also coached by a nationally accredited cycling coach. All participants provided written consent prior to any data collection and the study was approved by the Curtin University Human Research Ethics Committee (HRE2019-0418). 

Data collection occurred over a one-week period in August and September 2023 at an outdoor cycling track in Manning, Western Australia (Temperature: 20.6–29.8 °C, Humidity: 31.5–57.5%). The cycling track was an 800 m long loop including a 150 m straight section. The entire track was used for the 10 min warm-up but only the straight portion was to collect sprinting data. During the sprints, three-dimensional trunk, upper and lower limb kinematics were measured at 200 Hz via a set of wireless inertial measurement units (IMU) (Noraxon Myomotion, Scottsdale, AZ, USA) using previously reported placements [14,15]. Briefly, 15 sensors were attached to participants over the following locations: C7, T12, L5, pelvis and bilaterally on the hands, forearms, upper arms, thighs, shanks, and feet. Each sensor has an accelerometer of +/−200 g, a gyroscope of +/−7000°/s and a magnetometer of 16 gauss. The system was previously reported to have a valid evaluation of kinematics in team sports compared to traditional optical motion capture [15]. 

Upon arrival at the track, participants’ body mass (while wearing their standard cycling attire) and standing height were measured (using a Transtek BS-801-BT Body Scale and Craftright 30 m Tape Measure, respectively). Next, participants had IMUs attached to them, according to the manufacturer’s specifications, in the positions described. In addition to the IMU, a single global positioning system (GPS) (Catapult S5, Catapult Sports, Melbourne, Australia) collecting samples at 10 Hz was attached to each participant’s bicycle seat post to measure their velocity profile. All participants used their own bike, shoes, and cycling clothing so as not to interfere with the personalized set-ups cyclists normally adopt. We understand that allowing participants to use their own equipment introduces potential biases to the data; however, we endeavored to follow an ecologically valid approach, as all previous studies had only tested cyclists in a laboratory, merely making inferences to real-world cycling challenging. Following this, participants completed a 10 min warm-up at a self-selected effort with three 6 s familiarization sprints at minutes 6, 7, and 8 post-start [8,16]. 

After the warm-up, data collection commenced. First, participants were asked to simply cycle seated through the outdoor course at a constant speed of 20 km/h. This provided a baseline condition from which to compare kinematic changes during seated and standing sprinting. Participants completed the baseline condition once. Then, participants were recorded while sprinting within a 60 m “testing zone” which was set and marked on the 150 m straight section of the course. Prior to commencing the sprint, participants entered the testing zone at a speed of approximately 20 km/h, using their bicycle computer to monitor their speed. This procedure was completed six times, with the first three sprints completed standing (out of saddle) and the last three sprints seated (in saddle) [3]. After each sprint effort, participants were provided with a 3 min recovery period.

The GPS data were uploaded to manufacturer-supplied software (Catapult Sports, Melbourne, Australia). For each of the participants’ six trials, the maximum velocity achieved was determined for both in- and out-of-saddle conditions. Only the best trial (out of the three available) based on the maximum velocity as recorded by the GPS was analyzed further for kinematics. To determine this, three-dimensional kinematics were extracted from the IMUs using the manufacturer-supplied software (Noraxon myoResearch ver3.18, Scottsdale, AZ, USA). The kinematic data was sectioned into single pedal cycles for the middle eight revolutions of the 60 m seated, standing, and control conditions. An automatic algorithm detected the minimum knee flexion of the first revolution to the next minimum knee flexion of the next revolution. This was performed for both left and right knees and delineated all the kinematics into separate single cycles. Specific kinematics were selected and analyzed. This selection was based on previous work which emphasized the importance of the upper limb and trunk in sprinting [12] and included bilateral wrist flexion/extension, elbow flexion/extension, shoulder flexion/extension, and abduction/adduction. The truck kinematics analyzed included thoracic and lumbar spine flexion/extension, lateral flexion and axial rotation. Further, bilateral lower limb kinematics, including hip flexion/extension and abduction/adduction, knee flexion/extension, and ankle plantar/dorsiflexion, were also examined. Each cycle of the selected kinematics was then time-normalized to 0–100% of a cycle. The eight time-normalized revolutions for the selected kinematics were averaged to obtain an average (±SD) kinematic curve similar to those produced during typical running gait biomechanical analysis. Lastly, acceleration values for each participant were calculated using data recorded by the GPS.

The average, minimum and maximum joint excursion for the selected kinematics were obtained for the trunk and bilaterally for the upper and lower limbs. As this study was a series of three cases examining methods to obtain and process in-field cycling kinematic data, only descriptive statistics (mean and standard deviations (±SD)) were reported. 

## 3. Results

A lower peak velocity was observed during the in-saddle (Figure 1) conditions compared to out-of-saddle conditions (Figure 2). This was consistent across each of the participants. Participant 1 reached a peak velocity of 11.6 m/s at an acceleration of 0.7 m/s^2^ for standing sprinting and 10.4 m/s at an acceleration of 0.5 m/s^2^ for seated sprinting. Participant 2 reached 13.1 m/s at an acceleration of 0.8 m/s^2^ for standing sprinting and 11.7 m/s at an acceleration of 0.6 m/s^2^ for seated sprinting, while Participant 3 reached 13.9 m/s at an acceleration of 0.9 m/s^2^ during standing sprinting and 13.0 m/s at an acceleration of 0.9 m/s^2^ during seated sprinting.

Upper limb kinematic analysis revealed that participants maintained a stable joint angle through the pedal stoke during baseline cycling, with very small changes (≈5°) in their range of movement (Table 1 and Figure 3A,D,G,J,M). During seated sprinting, wave-like patterns started to emerge with increases in joint excursions (≈20°). When participants performed standing sprinting, discernable cyclic patterns emerged, especially for wrist flexion and extension (Figure 3C) and shoulder abduction and adduction (Figure 3O). Other joint kinematics did seem to display a wave-like pattern, but individual differences also emerged. For example, Participant 2’s elbow and shoulder flexion and extension displayed a different pattern to that of Participant 1 and 3 (Figure 3I,L). These differences occurred during the up phase of the pedal stroke for the elbow and throughout the whole stroke for the shoulder. Larger joint excursion (30–40°) was also noted in wrist and elbow flexion and extension and shoulder abduction and adduction. Shoulder flexion reduced by approximately 20° during standing sprinting compared to seated sprinting and baseline cycling. 

In general, participants adopted a flexed lumbar spine and an extended thoracic spine during baseline cycling (Table 1 and Figure 4). Further lateral flexion and axial rotation in the lumbar and thoracic spine were close to neutral during baseline cycling; however, there were individual differences (Figure 4A,D,G,J,M,P). Participant 2 adopted a posture of left laterally flexed thoracic spine and a right laterally flexed lumbar spine during baseline cycling (Figure 4D,M). Participants adopted approximately 5°–10° less lumbar flexion as they moved from baseline cycling through to standing sprinting (Figure 4J–L). However, Participants 1 and 3 adopted a more neutral thoracic spine posture while Participant 2 maintained an extended thoracic spine while sprinting in both seated and standing positions (Figure 4B,C). Small joint excursions (approximately 10° peak-to-peak) were recoded for thoracic and lumbar lateral flexion and axial rotation movements (Figure 4F,I). Thoracic axial rotation also displayed a wave-like pattern during the standing sprint (Figure 4I).

Hip and knee flexion and extension followed a very similar pattern for all participants in the three conditions tested (Table 1 and Figure 5A–C,G–I). Greater hip and knee extension (up to 10°) were recorded in all participants during the standing sprints (Table 1 and Figure 5C,I). Hip abduction and adduction and ankle dorsiflexion and plantar flexion were varied during baseline cycling (Figure 5D,J). These kinematics developed into more discernable, wave-like patterns when participants performed the seated and standing sprints (Figure 5E,F,K,L). 

## 4. Discussion

This study aimed to describe selected kinematics during road cycle sprints and to understand the influence that completing the sprint seated versus standing had on these kinematics. Further, a method of analyzing cycling kinematics similarly to running kinematics was trialed to judge the utility of this approach. We found that standing sprinting produced greater final velocity and acceleration compared to seated sprinting. The upper limb and trunk kinematics remained stable during baseline cycling but discernable patterns started to emerge during seated sprinting and were much more pronounced and obvious during standing sprinting. Lower limb kinematics tended to follow distinct patterns, especially sagittal plane movement in the hips and knees. Ankle moment showed more variation during baseline cycling but was more pattern-like during sprinting.

The velocity profiles observed from out-of-saddle and in-saddle sprinting differed in this study. There was a difference in final velocity for standing versus seated sprinting, which was consistent for all participants. It has been shown that greater power output occurs during standing sprint-cycling versus seated sprint-cycling [17,18], and this can explain the difference noted in the present study. Further, a previous biomechanical review described the relationship between increased leg-produced force, increased hip-reaction force, and decreased seat-reaction force during seated sprints and suggested cyclist compensate for this by pulling on the handlebars until they stand and sprint out of saddle [11]. Stone and Hull [19] suggested during standing sprinting, the hips are placed further forward compared to seated sprinting, creating greater crank arm-leverage, which explains the increased speed observed during this form of sprinting. The shoulder flexion kinematics from our study support this hypothesis, as decreased shoulder flexion was observed in all three participants during the standing sprint. Further investigations involving greater numbers of cyclists of all abilities are needed to confirm these findings.

Analysis of upper body kinematics showed that participants adopted a more stable posture during baseline cycling, with increasing ranges of movement as they sprinted. Holliday et al. [20] also showed larger average movement in the elbow during more intense (90% of max) compared to less intense cycling (60% of max). The average elbow flexion values reported by Holliday et al. [20] are similar to the values we obtained from participants during seated sprinting compared to higher levels of cycling intensity. These authors also reported little change in the shoulder flexion angle between the intensities tested, which is similar to our observations when our baseline condition was contrasted against seated sprinting. It was only during the standing sprint when shoulder flexion angles decreased in all participants. This can be explained by Stone and Hull’s [19] hypothesis that cyclists place their hips further forward during standing sprints. Only small shoulder abduction and adduction ranges were recorded in our study and there was variance in this angle between participants, especially during the baseline and seated sprint conditions. This can be attributed to the differences in handlebar width between our participants’ bicycles, especially considering that the difference in ranges and postures diminished during the standing sprint, where handlebar width may be less of a factor. Shoulder abduction and adduction movement may be an important consideration for bicycle fit and future studies may want to consider this movement in their research.

Concerning wrist joint excursions, although there were smaller ranges of movement in wrist flexion and extension as well as wrist radial and ulnar deviation during baseline cycling, these ranges increased during seated and standing sprinting. Further, differences between participants and differences between left and right wrists were also noted, especially for wrist radial and ulnar deviation. In fact, some of the values obtained for wrist radial deviation were potentially close to a full range of motion [21]. These findings are of interest, considering that radial deviation may be of more significance for cycling performance than previously considered. The wrist joint may play a similar role in sprint cycling performance to that which the ankle joint plays sprint running performance. Martín-Fuentes & van den Tillaar [22] found that the time from dorsal flexion to toe-off had a significant impact on performance among sprint runners. Future research describing the relationship between sprint cycling performance and time and radial deviation range should seek to determine if the wrist in sprint cycling is like the ankle in sprint running. If such an association is found, specific training can be considered for the improvement of sprint cycling performance, as ankle-specific training is already integrated in training for elite sprint running [23]. Wrist kinematics should be further investigated in a larger sample of track cyclists to elucidate the role the wrist plays in cycling sprinting.

The trunk kinematics recorded in our study showed that participants adopted a forward-flexed posture in the lumbar spine with smaller ranges in the frontal and transverse plane for both the lumbar and thoracic spine. Participants’ forward flexed posture in the lumbar spine decreased during sprinting. Participants also adopted an extended posture in the thoracic spine, and this extension also decreased as participants sprinted. Our lumbar spine results are similar to those previously reported, but our thoracic spine results differ substantially from those of other researchers [20]. The differences can be attributed to two main factors. Firstly, Holliday et al. [20] calculated and reported thoracic spine orientation angle relative to the laboratory coordinate system while our thoracic spine is reported relative to the lumbar spine. Our method has been shown to be a valid approach to understanding spinal kinematics in fast bowling during cricket games [24]. Secondly, Holliday et al. [20] tested participants using an ergometer in a laboratory, while our data were obtained during field-based cycling. It may be that during cycling, it is paramount to have an extended thorax for forward gaze to allow cyclist to gauge road conditions, maintain balance, and avoid obstacles. Frontal and transverse plane kinematics were smaller in magnitude and range compared to those recorded in the sagittal plane. However, substantial differences were recorded between participants and cycling conditions. For example, Participant 2 adopted a somewhat scoliotic posture during baseline cycling with pronounced left lateral flexion in the thoracic spine offset by right lateral flexion in the lumbar spine. This posture did not manifest during seated or standing sprinting, in which a more neutral spine posture was adopted. More research is needed to further understand trunk kinematics in cycling.

The lower limb kinematics in our study demonstrate a distinct pattern of movement, especially in the hips and knees in the sagittal plane during all conditions, the hips in the frontal plane, and the ankle in the sagittal plane during the sprinting conditions. Our results for hip flexion and extension, hip abduction and adduction, and knee flexion and extension during baseline cycling show similar patterns to those reported by Yum et al. [25]. However, the magnitudes of the ranges of motion are larger in our study. This may be attributed to the fact our participants rode at 20 km/h during baseline cycling while Yum et al. [25] had their participants ride at 10–12 km/h on an ergometer. In contrast, our results recorded during seated sprinting for knee extension at bottom dead center are similar in magnitude to those found in the work of Holliday et al., [20]. However, our results for hip extension at top dead-center are lower compared to these authors’ results. Kinematics derived from IMUs have been shown to be very similar to optical motion capture data, especially in the knee, but variances have been recorded for hip kinematics, and this may, in part, explain the discrepancy in results.

Our results for ankle dorsi and plantar flexion during baseline cycling differed from those of Yum et al. [25], with our participants adopting a more plantarflexed posture. Our participants used clipless pedals to attach their shoes to the bicycle, while pictures of Yum et al.’s [25] configuration suggested their participants rode barefoot on an ergometer. The use of clipless pedals allows cyclists to exert pull forces during the upward phase of the pedal stroke and, as such, allows them to exhibit a more plantarflexed foot. Our results also show that participants adopted more knee extension and ankle dorsiflexion range as they sprinted. These results are in agreement with the results of others who have also showed these changes with incremental cycling intensities. However, our results show little change in the hip extension range, while others have shown increases. Our participants needed to adopt postures that allowed forward vision, while the other researchers cited performed their experiments in a laboratory where forward vision was not prioritized. 

A further confounder of comparison with the available literature is the potential differences between ergometer frame stiffness and actual bike frame stiffness characteristics and how these can influence cyclists’ kinematics. During seated cycling, the energy produced by muscles is delivered to three points: pedals, seat, and handlebars [11]. Baker and Davies [10] reported the importance of upper limb function during high-intensity cycling and the role of the handlebar; this was conducted on a relatively stable laboratory ergometer. Building on this finding, Costes et al. [9] highlighted the upper limb energy-transfer contributions (3–5%) to total crank power output during real-world cycling, but found that this is compromised if upward forces are not produced on the handlebar. Turpin et al. [12] hypothesized how upper limb joints stabilize the trunk to create seat force and transfer energy down to the lower limb. While our current data do not provide insight on how the forces produced by the legs interact with seat reaction force, we do provide initial evidence on the truck kinematics and how these seek to compensate for system stiffness differences between seated and standing sprinting and the handlebar interaction. 

Our method of collecting, analyzing, and reporting cycling kinematics has shown that field-based cycling kinematics do vary from those observed during laboratory-based studies and that extrapolating results from the laboratory may be problematic. Further, the characterization of whole pedal stroke data, like the approach used in running gait analysis, shows promise, as these data are rich and insightful. Features such as movement variability as well as bilateral differences during the whole pedal stroke can be analyzed. Further, subtle differences between cyclists can also be studied. This may improve our understanding of this popular sport and may lead to performance enhancements, injury prevention, and optimal rehabilitation and return to cycling after injury. It may also give us insights into errors made by cyclists, which can result in serious crashes and injury.

Our study is limited by the small sample size and, thus, the ability to extrapolate the findings to a broader population. Consistent findings across a larger sample of cyclists are needed to improve the level of evidence. Furthermore, our participants are triathletes, and although they were familiar with criterium racing as well as being coached by a nationally accredited cycling coach, potential inclusion of varying levels of road cyclists and/or track cyclists is needed. This approach will ensure a rich data source for the optimization of cycling. Allowing participants to use their own equipment can easily add to the variance obtained in our kinematics. Further, environmental factors such as wind cannot be controlled or accounted for. Our approach, however, allows scientists interested in cycling to study these phenomena, allowing us to expand our knowledge about cycling.

## 5. Conclusions

Our study showed that our participants maintained a stable posture in most joints studied during baseline cycling, but substantial changes in kinematics were noted as they performed seated and standing sprints. In particular, discernable patterns started to emerge in the upper limb joints and the ankle. Specific postures in the trunk were maintained depending on the cycling activity. Our approach also showed that although the patterns of kinematics in many joints were similar to those reported in previous laboratory studies, the magnitude of the ranges of movement do differ. Our approach also highlighted insightful results where movement variability within and between cyclists can be studied. Future research should examine the in-field kinematics of a larger sample of road and track cyclists during sprinting. This could facilitate a definitive examination of the association of certain kinematic strategies with a superior sprinting performance. This information will greatly enhance coaching and training strategies in competitive cycling. 

## Figures and Tables

**Figure 1 sensors-24-03453-f001:**
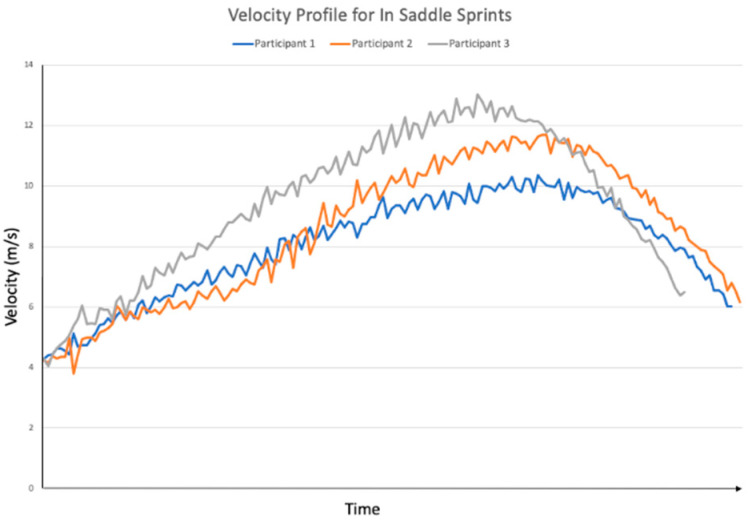
Velocity over time for each participant during the seated sprint condition.

**Figure 2 sensors-24-03453-f002:**
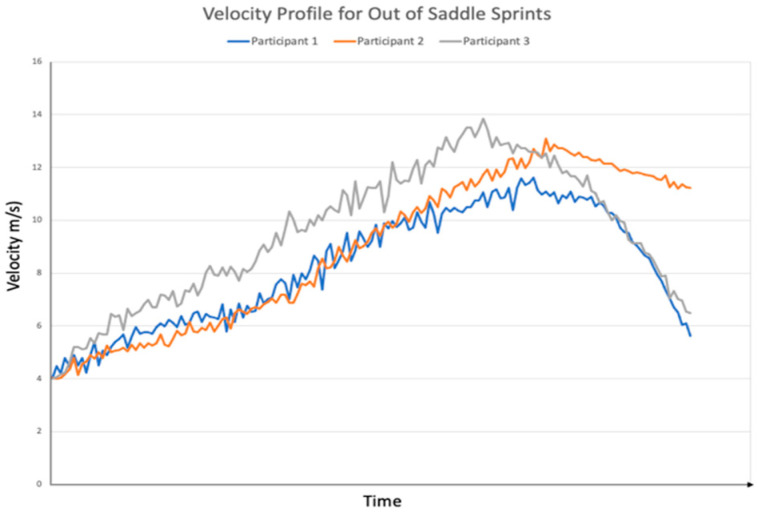
Velocity over time for each participant during the standing sprint condition.

**Figure 3 sensors-24-03453-f003:**
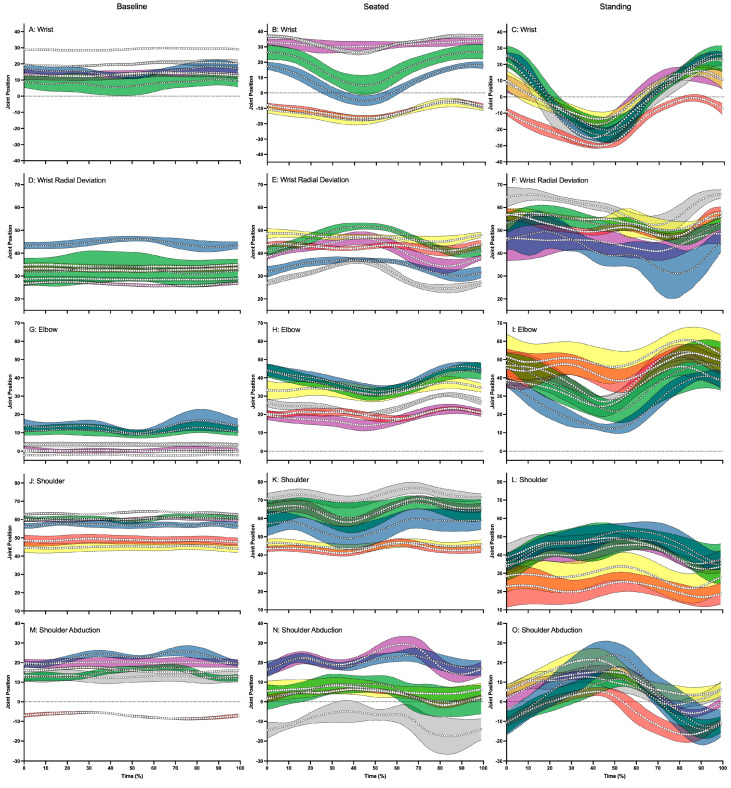
This figure illustrates the wrist flexion (positive) and extension (negative) (**A**–**C**), wrist radial (positive) and ulnar (negative) deviation (**D**–**F**), elbow flexion (positive) and extension (negative) (**G**–**I**), shoulder flexion (positive) and extension (negative) (**J**–**L**) and shoulder abduction (positive) and adduction (negative) (**M**–**O**) average joint positions and standard deviation across baseline, seated and standing cycling conditions for Participant 1 (△: blue—left and green—right), Participant 2 (☐: red—left and yellow—right), Participant 3 (◯: gray—left and purple—right). In each panel where applicable the zero (0) has been shown with a dotted line.

**Figure 4 sensors-24-03453-f004:**
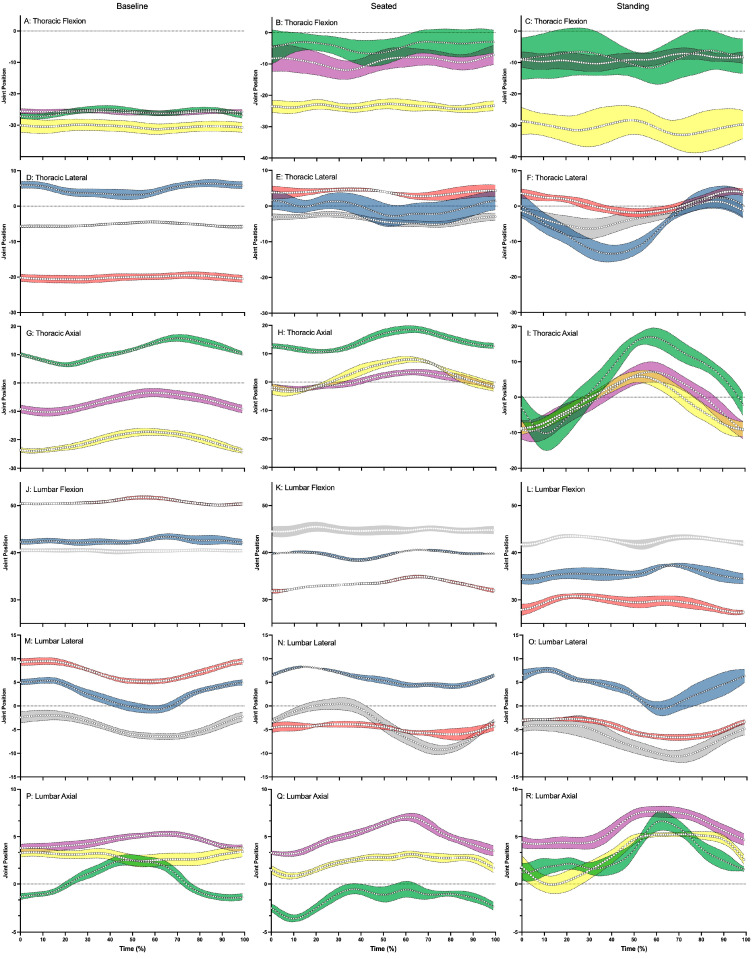
This figure illustrates the thoracic flexion (positive) and extension (negative) (**A**–**C**), thoracic lateral flexion (positive = right lateral flexion, negative = left lateral flexion) (**D**–**F**), thoracic axial rotation (positive = right axial rotation, negative = left axial rotation) (**G**–**I**), lumbar flexion (positive) and extension (negative) (**J**–**L**) lumbar lateral flexion (positive = right lateral flexion, negative = left lateral flexion) (**M**–**O**), lumbar axial rotation (positive = right axial rotation, negative = left axial rotation) (**P**–**R**) positions and standard deviation across baseline, seated, and standing cycling conditions for Participant 1 (△: blue—lateral and green—axial), Participant 2 (☐: red—lateral and yellow—axial), Participant 3 (◯: gray—lateral and purple—axial). In each panel where applicable the zero (0) has been shown with a dotted line.

**Figure 5 sensors-24-03453-f005:**
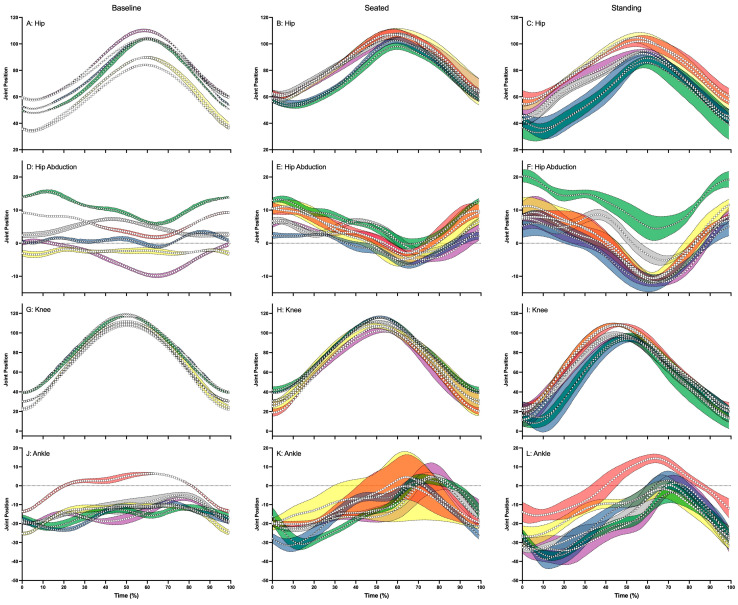
Illustrates the hip flexion (positive) and extension (negative) (**A**–**C**), hip abduction (positive) and adduction (negative) (**D**–**F**), knee flexion (positive) and extension (negative) (**G**–**I**) and ankle dorsiflexion (positive) and plantarflexion (negative) (**J**–**L**) average joint positions and standard deviation across baseline, seated and standing cycling conditions for Participant 1 (△: blue—left and green—right), Participant 2 (☐: red—left and yellow—right), Participant 3 (◯: gray—left and purple—right). In each panel where applicable the zero (0) has been shown with a dotted line.

**Table 1 sensors-24-03453-t001:** Avatars of typical postures adopted by participants for each condition. Near-top dead center (left leg)/bottom dead center (right leg) are depicted. Please note that as the head position was not measured, the software has assumed a neutral head position was maintained relative to the spinal column and grayed the head and cervical spine.

	Baseline	Seated	Standing
Participant 1	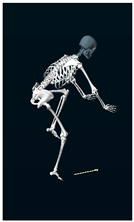	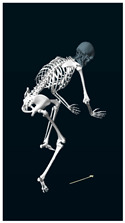	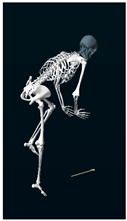
Participant 2	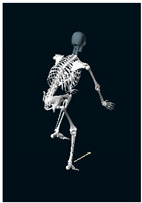	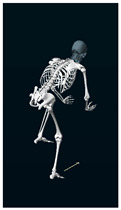	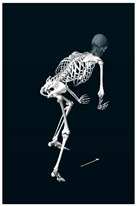
Participant 3	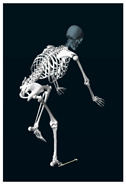	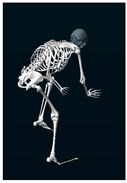	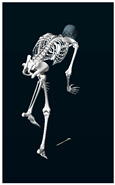

## Data Availability

Raw kinematic data from this study can be obtained by contacting the corresponding author.

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
