# Peer review of "Using Inertial Measurement Units to Examine Selected Joint Kinematics in a Road Cycling Sprint: A Series of Single Cases"

_sensors, 2024, doi:10.3390/s24113453_

Round 1

Reviewer 1 Report

Comments and Suggestions for Authors

The paper investigates upper limb kinematics during road cycling sprints. While the background and justification for this investigation are sound, the study's approach requires further refinement. The small sample size is acknowledged, but the absence of crucial kinematic variables, such as shoulder, trunk, and lower limb motions, hinders the discussion of results. The study focuses solely on elbow and wrist motions, which require robust justification, and fails to communicate the absence of shoulder motion in the methods section, despite its significant influence on elbow and wrist movements. Trunk inclination is mentioned in the discussion without corresponding results, and the potential impact of lower limb kinematics on upper limb movements is not addressed.

Addressing the small sample size is indeed important, but it is suggested that the study aims for a more comprehensive analysis rather than holding back data for future submissions. The selection of only elbow flexion/extension, wrist flexion/extension, and radial/ulnar deviation needs robust justification. Additionally, the absence of shoulder motion should be clearly communicated in the methods, as shoulder movements significantly influence elbow and wrist motion.

The impact of trunk inclination on elbow kinematics is mentioned in the discussion but lacks corresponding results. Given the placement of inertial measurement units (IMUs), including trunk kinematics is crucial and should be reported. Although the study focuses on upper limb kinematics, the discussion also touches upon lower limb force production and hip position. Therefore, including lower limb kinematics would provide a more comprehensive understanding of potential confounding factors in the relationship between upper limb kinematics and acceleration.

Figures could be improved for clarity, such as combining Figures 3 and 4 into one figure and presenting plots side by side for clearer comparison. The same approach could be taken with Figures 5 and 6, as well as 7 and 8. This would allow for a clearer depiction of how kinematics vary throughout the pedal stroke and potentially enable the inclusion of other relevant kinematic variables.

Suggestions:

Provide robust justification for the selection of elbow and wrist motions as the focus of the study and clearly communicate the absence of shoulder motion in the methods section.

How do you justify focusing solely on elbow and wrist motions while neglecting other key kinematic variables such as shoulder, trunk, and lower limb motions?

Could you clarify why shoulder motion was not included in the analysis and ensure that this omission is clearly communicated in the methods section?

Report trunk kinematics, as they are crucial for understanding upper limb movements and could play a significant role in the study's findings.

Address the potential impact of lower limb kinematics on upper limb movements to ensure a comprehensive analysis of confounding factors.

Overall, enhancing the clarity and completeness of the analysis would strengthen the study's findings and implications.

Author Response

Reviewer 1

The paper investigates upper limb kinematics during road cycling sprints. While the background and justification for this investigation are sound, the study's approach requires further refinement. The small sample size is acknowledged, but the absence of crucial kinematic variables, such as shoulder, trunk, and lower limb motions, hinders the discussion of results. The study focuses solely on elbow and wrist motions, which require robust justification, and fails to communicate the absence of shoulder motion in the methods section, despite its significant influence on elbow and wrist movements. Trunk inclination is mentioned in the discussion without corresponding results, and the potential impact of lower limb kinematics on upper limb movements is not addressed.

Addressing the small sample size is indeed important, but it is suggested that the study aims for a more comprehensive analysis rather than holding back data for future submissions. The selection of only elbow flexion/extension, wrist flexion/extension, and radial/ulnar deviation needs robust justification. Additionally, the absence of shoulder motion should be clearly communicated in the methods, as shoulder movements significantly influence elbow and wrist motion.

The impact of trunk inclination on elbow kinematics is mentioned in the discussion but lacks corresponding results. Given the placement of inertial measurement units (IMUs), including trunk kinematics is crucial and should be reported. Although the study focuses on upper limb kinematics, the discussion also touches upon lower limb force production and hip position. Therefore, including lower limb kinematics would provide a more comprehensive understanding of potential confounding factors in the relationship between upper limb kinematics and acceleration.

Figures could be improved for clarity, such as combining Figures 3 and 4 into one figure and presenting plots side by side for clearer comparison. The same approach could be taken with Figures 5 and 6, as well as 7 and 8. This would allow for a clearer depiction of how kinematics vary throughout the pedal stroke and potentially enable the inclusion of other relevant kinematic variables.

Response

Thank you for your comments and we have endeavoured to address your suggestions in a point-by-point format below. We have included these as changes in the manuscript and highlighted these.

Suggestions:

Provide robust justification for the selection of elbow and wrist motions as the focus of the study and clearly communicate the absence of shoulder motion in the methods section.

We have performed a comprehensive re-analysis of the data and have now included kinematic results from the wrist, elbow, shoulder, trunk (thoracic and lumbar), hip, knee and ankle. We have also included a baseline condition, cycling at 20km/h as a condition for comparison.

How do you justify focusing solely on elbow and wrist motions while neglecting other key kinematic variables such as shoulder, trunk, and lower limb motions?

We thank the reviewer for these comments. We have now included results for the upper and lower limbs as well as the trunk. We have processed the data in a similar fashion to running gait ie in an average (±SD) waveform, over eight pedal strokes, from minimum knee extension to the following minimum knee extension.

Could you clarify why shoulder motion was not included in the analysis and ensure that this omission is clearly communicated in the methods section?

We thank the reviewer for this point. We have now included shoulder flexion/extension and abduction/adduction kinematics for each of the conditions.

Report trunk kinematics, as they are crucial for understanding upper limb movements and could play a significant role in the study's findings.

We agree and we have now included three-dimensional kinematics for the thoracic and lumbar spine. We have also included in the Discussion, points about these findings.

Address the potential impact of lower limb kinematics on upper limb movements to ensure a comprehensive analysis of confounding factors.

We have included lower limb kinematics in the Results and Discussion. We have highlighted the changes in the revised manuscript.

Overall, enhancing the clarity and completeness of the analysis would strengthen the study's findings and implications.

We have included a completely revised Discussion section that address clarity and completeness issues. These are highlighted in the revised manuscript.

Reviewer 2 Report

Comments and Suggestions for Authors

Originality of the text 76.57%.

The obtained data are not statistically significant (unreliable). Three people for serious research is very little. When studying kinematic characteristics, graphics should be presented in three-dimensional form. The data obtained do not have any theoretical or practical significance. I didn't see the hypothesis in the article. I believe that the article cannot be published in this form.

Author Response

Reviewer 2

The obtained data are not statistically significant (unreliable). Three people for serious research is very little. When studying kinematic characteristics, graphics should be presented in three-dimensional form. The data obtained do not have any theoretical or practical significance. I didn’t see the hypothesis in the article. I believe that the article cannot be published in this form.

Thank you for your comments. We have revised the manuscript to not include any inferential statistics. Instead, we decided to strengthen the case series narrative by including individual level results. We also took the opportunity to investigate a novel approach to analysing road cycling data in a way that has not been performed before. We have highlighted the changes in the revised manuscript.

Reviewer 3 Report

Comments and Suggestions for Authors

Check the attached file.

Author Response

Reviewer 3

The study titled "Using Inertial Measurement Units to Examine the Relationship between Upper Limb Joint Kinematics and Acceleration in a Road Cycling Sprint: A Series of Single Cases" contributes to understanding the biomechanics of road cycling sprints outside the laboratory setting. It focuses on upper limb kinematics, comparing in-saddle and out-of-saddle sprinting, and revealing two distinct acceleration phases in both sprint types. The findings suggest variability in kinematic strategies among cyclists and highlight the potential for personalized training approaches. Despite its insightful contributions, the study's small sample size and focus on triathletes as participants point towards the need for further research with a broader and more varied cyclist population to generalize these findings. This work lays a promising foundation for future investigations into cycling kinematics, potentially influencing training, performance strategies, and equipment design in competitive cycling. However, there is margin for improving the manuscript:

  • Small Sample Size: The study is limited by its small sample size (only three participants), which could affect the generalizability of its findings. A larger sample would provide more robust data and allow for more confident conclusions to be drawn about the cycling population at large. Elaborate in the justification of the selection of the sample size.

Thank you for this comment and we do acknowledge the small sample size. We have now removed the inferential statistics as we felt try to generalise any inferences to a larger community would be meaningless. As such, we have refocused the paper to highlight an approach rather than making comparisons. The majority of the Results section is now rewritten and highlighted in yellow in the revised manuscript.

  • Participant Selection: All participants are highly trained/national level triathletes. This specific athlete population may have unique biomechanics compared to road or track cyclists, which could limit the applicability of the findings to broader cycling disciplines.

We acknowledge this sample bias in the limitations section as well as clarify some of the participants characteristics in the Methods section.

  • Statistical Significance Threshold: The choice of a statistical significance level of p<0.1 is unconventional and may increase the risk of type I errors (false positives). This choice is justified by the small sample size and the nature of single-case research, but it does raise questions about the robustness of the findings. Please elaborate more on this selection.

We have omitted all inferential statistical testing as we felt making inferences for generalisability from such a small sample is fraught with challenges. Thus we have only reported individual cases as descriptive statistics. This is displayed in the revised Results and Discussion sections and highlighted in yellow.

  • Lack of Control for External Variables: While the study benefits from an environmentally valid setting by conducting tests outdoors, this approach may introduce variability due to external factors such as wind resistance, temperature, and track conditions. These factors are not controlled for or discussed in detail, which could influence the kinematics and performance outcomes. Please discuss them with more detail.

We have included data from a baseline condition collected on the same day. We have also made comparisons to recently published laboratory data which gives us confidence in our results. We also acknowledge the limitations of not being able to control the environment. These changes are highlighted in the revised manuscript.

  • Limited Exploration of Kinematic Strategies: The study identifies variability in kinematic strategies among participants but does not deeply explore the implications of these strategies for performance or injury prevention. Further analysis could enhance understanding and application of the findings.

We have now included selected whole-body kinematics in the analysis. We have also incorporated a discussion of performance-based characteristics in the revised Results and Discussion section. These are highlighted in the revised manuscript.

  • The study exclusively investigates upper limb kinematics, overlooking the role of lower limbs and trunk, which are also critical in cycling performance. Including these areas in future research could provide a comprehensive understanding of the biomechanics involved in cycling.

We have included lower body and trunk kinematics in the revised manuscript. This has given a much deeper insight to the data. Please see highlighted sections in the revised manuscript.

  • Potential for Equipment Variation: Participants used their own bikes, shoes, and cycling attire, which introduces variability in equipment that could affect kinematics and performance. While this adds ecological validity, it also complicates the ability to isolate the effects of body movements from those of equipment differences.

We acknowledge that this is a limitation of our approach. In fac, some of our new, included data suggests differences in handlebar width may have caused some of the differences in kinematics between participants. We have added statements to acknowledge this in our revised manuscript. Please see the highlighted sections.

  • Please enhance the quality of the plots to improve readability.

We have removed the old graphics and included higher quality images of our new results.

Given the insightful contributions and identified weaknesses of the manuscript, my overall recommendation would be to accept the manuscript after revisions. The authors should consider expanding the discussion section to address the limitations explicitly, such as the small sample size and the specific demographic of participants (triathletes). Additionally, a deeper exploration of the implications of their findings for training and performance strategies and injury prevention could enrich the manuscript's value to the reader.

We have included in our new Discussion section, limitations to our sample as well as implication of this data for training and performance.

Round 2

Reviewer 1 Report

Comments and Suggestions for Authors

The authors have sufficiently addressed the issues raised in the previous review.